# Dynamic marine viral infections and major contribution to photosynthetic processes shown by spatiotemporal picoplankton metatranscriptomes

Ella T. Sieradzki [1], J. Cesar Ignacio-Espinoza[1], David M. Needham[1], Erin B. Fichot[1] & Jed A. Fuhrman [1]

Viruses provide top-down control on microbial communities, yet their direct study in natural environments was hindered by culture limitations. The advance of bioinformatics enables cultivation-independent study of viruses. Many studies assemble new viral genomes and study viral diversity using marker genes from free viruses. Here we use cellular metatranscriptomics to study active community-wide viral infections. Recruitment to viral contigs allows tracking infection dynamics over time and space. Our assemblies represent viral populations, but appear biased towards low diversity viral taxa. Tracking relatives of published T4-like cyanophages and pelagiphages reveals high genomic continuity. We determine potential hosts by matching dynamics of infection with abundance of particular microbial taxa. Finally, we quantify the relative contribution of cyanobacteria and viruses to photosystem-II *psbA* (reaction center) expression in our study sites. We show sometimes >50% of all cyanobacterial+viral *psbA* expression is of viral origin, highlighting the contribution of viruses to photosynthesis and oxygen production.

[1] Department of Biological Sciences, University of Southern California, 3616 Trousdale Pkwy, Los Angeles 90089 CA, USA. Correspondence and requests for materials should be addressed to J.A.F. (email: Fuhrman@usc.edu)

Marine viruses are an important component of biogeochemical cycles in the ocean, both as a top-down control on microbial populations[1–5] and as converters of particulate organic carbon to dissolved organic carbon by cell lysis, termed the viral shunt[4,6,7]. Viruses that infect bacteria, or phages, have also been shown to promote genomic diversity in their hosts[8–12]. Until recently, the bulk of the research on marine viruses focused on phages that infect cyanobacteria (cyanophages) that include T4-like myoviruses, T7-like podoviruses, and siphoviruses[13–19]. However, cultivation of hosts and mining of metagenomic and metaviromic datasets revealed abundant phages infecting heterotrophic hosts[20–25], some of which even have demonstrated diel cycles[26].

The lack of a universal marker gene for phages, in concert with extremely limited reference sequences with which to study viral metagenomes, made it much more difficult to study them compared to bacteria and archaea; thus, many studies instead focused on general enumeration and estimation of late-infection rates via microscopy, tracking phages with cultivable hosts, or studying marker genes within major phage groups (e.g. myoviruses using the *g20* or *g23* genes)[27,28]. While the addition of more cultured phage genomes to public databases enhances our understanding of these fascinating organisms[29], comprehensive analysis would ultimately require more knowledge on culture conditions for hosts, and the actual cultivation of hundreds to thousands of potential hosts, all very challenging. The rise of next-generation sequencing and metagenomics made the study of environmental phages more accessible[6,26,30–32]. In recent years new techniques for the study of viruses, such as vSAGs, polonies, and viral-BONCAT, were introduced, and promise to move the field further into the realm of culture-independent research[31,33–35].

Most cyanomyoviruses and some cyanopodoviruses have been shown to carry horizontally transferred auxiliary metabolic genes (AMGs) that presumably maintain the photosynthetic machinery functional during infection, and may even divert energy into nucleotide production instead of carbon fixation[36–43]. Viral *psbA* genes coding for Photosystem II (PS-II) photosynthetic reaction center protein D1 have been shown to be widespread and discernable from the host versions[40,44]. This protein has a short lifetime, and if not replenished, PS-II function fails[37,41,45]. Because viruses generally shut down synthesis of new mRNA coded by the host genome, expression of the viral version of this gene is required to maintain photosynthesis, and it is expressed almost throughout the cyanophage latent period[8,40].

We have recently become particularly interested in direct evidence of active viral infections. During lytic infection of bacterial hosts, viral genes are generally thought to be expressed as polycistronic mRNA corresponding to sequential phases of infection[46–49]. Thus, active viral infections in mixed natural communities could be tracked by assembling and characterizing multigene transcripts from cellular microbial metatranscriptomes. While some such transcripts may represent host operons or other vectors, recently developed methods can reasonably well identify sequences of viral origin[50,51]. Here we use that approach to characterize active viral infections via marine metatranscriptomic and metagenomic analyses and show temporal and spatial patterns in the diversity and activity of such infections.

Here we study surface seawater from different seasons over three sites across the San Pedro Channel, Southern California, USA: The Port of Los Angeles (POLA), Santa Catalina Island Two Harbors (CAT), and the San Pedro Ocean Time-series (SPOT). These sites are within a transect of 37 km and represent a gradient of human impact with POLA being the most impacted and SPOT resembling open ocean conditions[52]. At all three sites, free virus-like particles, counted in whole seawater, outnumber bacteria and archaea roughly 10:1 (Supplementary Figure 1). For metagenomic and metatranscriptomic analyses, we examine only the 0.2–1 µm size-fraction, which includes most free-living bacteria, archaea, and some picoeukaryotes.

We show that most assembled viruses in our dataset appear only ephemerally, have no close relatives and most of them represent phages infecting heterotrophic bacteria. Read recruitment to curated cyanophage genomes indicates presence of many close relatives creating a pattern of genomic continuity. Host and phage spatiotemporal dynamics are used to match an assembled cyanophage with a *Synechococcus* strain. Finally, we reveal a high contribution of cyanophages to gene abundance and expression of photosystem-II.

## Results

**Spatiotemporal patterns of active infections.** Via assembly of metatranscriptomes, we obtained 1455 contigs longer than 5 kbp, of which 61 (3.9%) were characterized as viral using Virsorter and VirFinder (see methods and Supplementary Data 1). Additionally, a cross-assembly of the metatranscriptomic viral contigs with metagenomes of the same samples ($n = 12$) yielded nine more contigs (mean length 26,563 bp) characterized as viral. Most of the contigs represent dsDNA viruses ($n = 68$) as apparent from their presence in metagenomes (evaluated by read recruitment, see methods), but one appeared to be an RNA virus possibly infecting a eukaryotic host. This contig contained an RNA-dependent-RNA-polymerase whose nearest match in NCBI non-redundant database (NR) was marine Antarctic phytoplankton RNA virus PAL_E4[53].

These 69 viral contigs revealed varied patterns of presence (in metagenomes) and activity (in metatranscriptomes) in the three sites over a year (Fig. 1). Some regional patterns were evident, e.g. some viral contigs were unique to the Port of LA (Fig. 1), and POLA always differed from the other sites in expression of viral contigs (by Bray–Curtis dissimilarity) more than SPOT and CAT differed from each other (Supplementary Figure 2B). This pattern corresponds to the difference in microbial abundance and heterotrophic production between the port and the other sites (Supplementary Figure 1), and to patterns of microbial community composition and activity by amplicon single variants of 16S-rRNA (ASV, Supplementary Figure 2C, D).

We did note that for most sites and dates, the recruitment patterns in metatranscriptomes (presumably mRNA from viruses within the latent period) very generally reflected the metagenomic recruitment (presumably DNA in late infection and assembled viruses, when the highest amount of viral DNA is present in cells), with notable exceptions of POLA and SPOT in April and CAT in January, where the most transcriptionally active contigs had relatively low DNA as shown by recruitment in the metagenomes (Fig. 1).

Persistent infections (mean coverage $>= 0.75x$ in at least three out of four metatranscriptomes per site), comprising seven out of 69 contigs, were found mostly at CAT and SPOT (Supplementary Data 1). Two contigs were persistent in all three sites, and none were persistent only at POLA. Non-persistent, i.e. ephemeral, infections dominated the assembled landscape, as 62 out of 69 of the contigs only appeared in 0–2 out of four dates per site. Bray–Curtis dissimilarity of the relative abundance of viral contigs presence (DNA) and expression (RNA) within each site as well as between sites was almost exclusively between 70–100%. High community-level dissimilarity indicates that even within site, most viral infections which we detected as instantaneous snapshots at the time of sampling, were localized and sporadic on the time scale we studied (Supplementary Figure 2A, B).

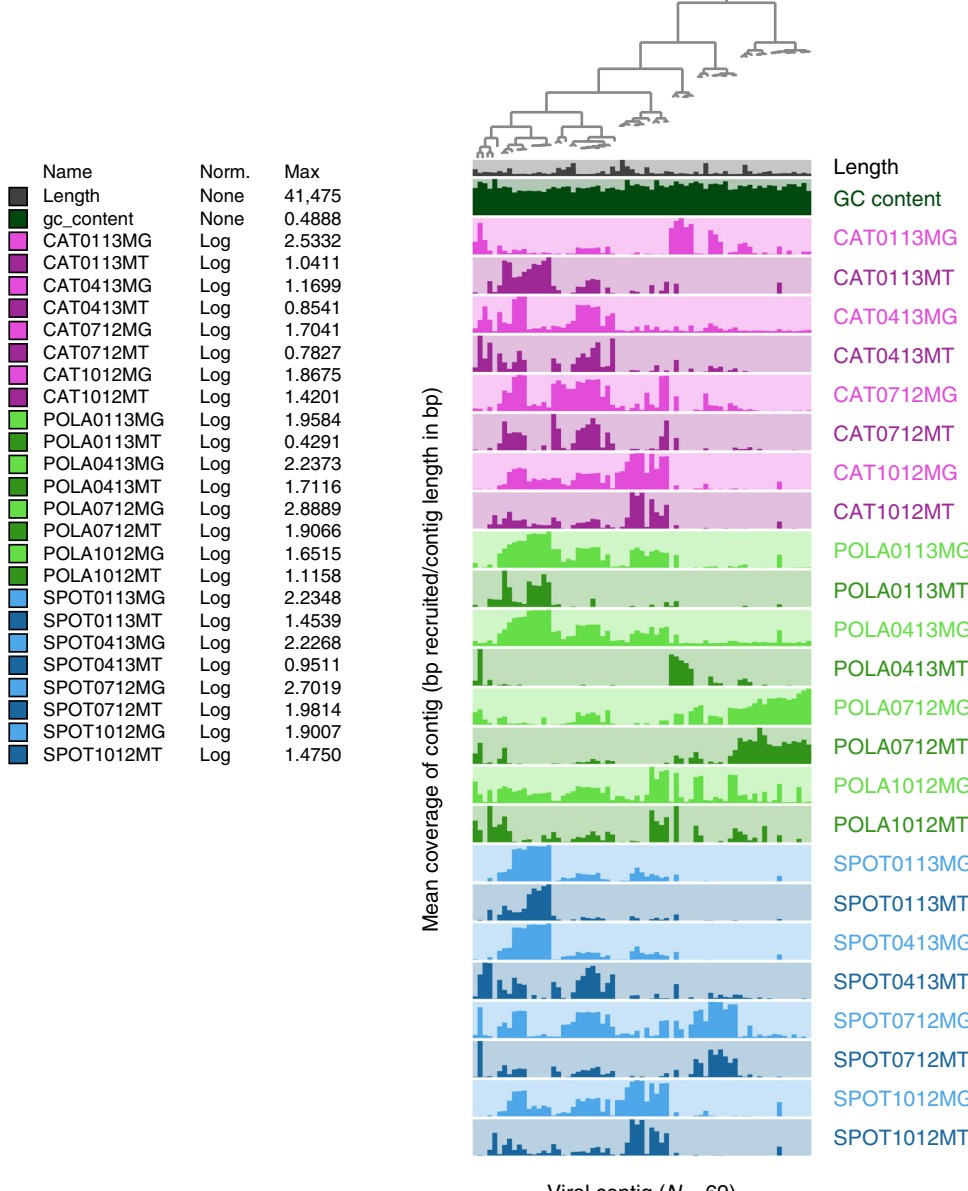

**Fig. 1** Dynamics of active infections over seasons and sites. Mean coverage, as a proxy of relative abundance, of 69 viral contigs across three sites (Port of LA—POLA, San Pedro Ocean Time-series—SPOT and Two Harbors—CAT) and four seasons (July 2012, October 2012, January 2013, and April 2013) in metagenomes (MG) and metatranscriptomes (MT). Each column represents one contig and each row represents one sample. The bar heights are relative to the highest log-transformed mean coverage within each sample, detailed in the legend column Max. Mean coverage, based on bowtie2 mapping, was calculated excluding contig positions whose coverage depth was in the top 25% of coverage, as they may be biased by non-specific recruitment localized to short, highly conserved portions of the contig (Supplementary Figure 3). The dendrogram denotes hierarchical clustering of the contigs by abundance[79]

**Genomic diversity of actively infecting phages**. To investigate the diversity of phages in our system, we mapped reads at varying identity percentages to the assembled active viral contigs as well as to published, curated cyanophage and pelagiphage genomes (see Supplementary Data 1 for accession numbers). The assembled viral contigs appeared to be biased toward low population-level microdiversity (i.e. more clonal) viruses (Fig. 2a, c). All the viral contigs we assembled in this study appear to have many nearly identical relatives but few moderately close ones as shown by recruitment plots (most recruitment at 98–100% identity and little recruitment at 90–97%, Fig. 2c). Read recruitment to published genomes of cultured pelagiphages was continuous along most of the genome (similar to Fig. 2a) at up to 100% identity (Fig. 2e) with high mean coverage (Supplementary Data 1), yet we

did not assemble any moderately-complete pelagiphage genome as determined by ORF annotations (Supplementary Data 2). In contrast, the published cyanophage genomes had little or no exact read recruitment in our samples, but rather recruitment centering broadly around 85–90% identity, suggesting cyanophages in our region are related to the cultured ones but are not identical (Fig. 2d). In fact, only three of our assembled contigs were recognizable as putative cyanophages, despite the ease in identifying cyanophages by similarity to known genomes or the presence of photosynthesis-related genes. Only one contig represented a potential partial pelagiphage genome. The low rate of success in assembling these phage types was surprising considering that their hosts—cyanobacteria and the SAR11 clade—represented a significant fraction of the microbial community in

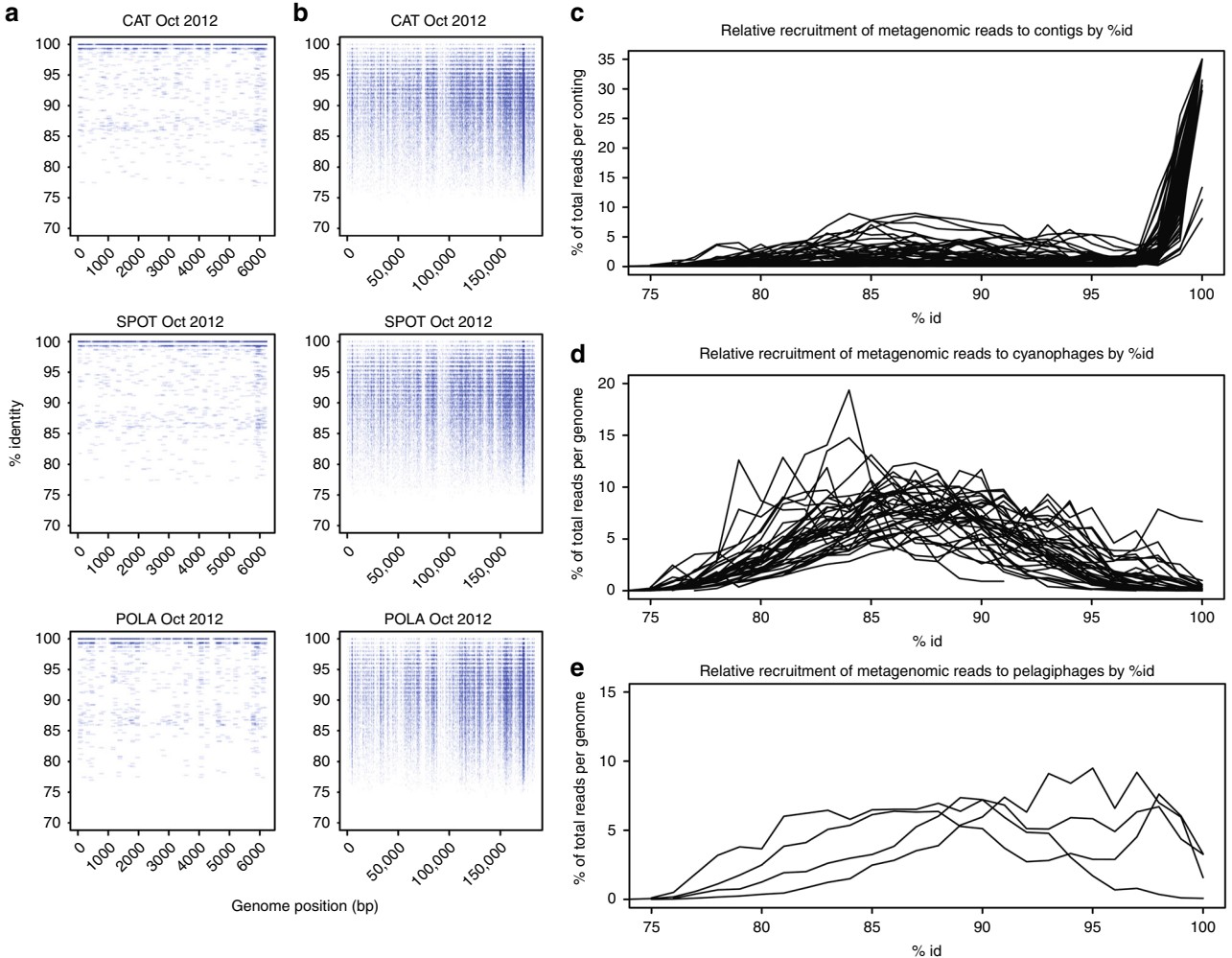

**Fig. 2** Genomic diversity of assembled and curated phages. Metagenomic read recruitment (minimum 100 bp hit length) from October samples (single sample per site) to **a** an assembled cyanophage contig, and **b** *Prochlorococcus* phage P-HM2 genome. The highest recruitment to the assembled contig in **a** was at 99–100% identity (high density near 100% is not fully evident from the graph due to overlaps, see **c**), whereas P-HM2 revealed a genomic continuum ranging from 80% to ~98% identity. **c–e** Recruitment as a function of percent identity of reads to assembled contigs (**c**), curated cyanophages (**d**) and curated pelagiphages (**e**) demonstrated that assembled contigs mostly recruited at 100% ID and had few moderately close relatives (**c**) whereas published reference genomes of 45 cyanophages revealed moderately close relatives of those genomes but few matches near 100% (**d**), and four pelagiphages had continuous recruitment implying a much less discrete spectrum of diversity (**e**). See Supplementary Data 1 for accession numbers of reference genomes

all sites at any given time, although more so at SPOT and CAT (Supplementary Figure 4). We recognize that the recruitment to contigs depends on how successful we were at assembling viral contigs, and to see if we could obtain contigs of cyanophage or pelagiphage by more sophisticated assembly methods, we also tried applying a subsampling-based assembly approach (see methods) on two samples: SPOT from October, which had the highest relative abundance of cyanobacteria, and POLA from April, which had the highest relative abundance of viral *psbA* (see below). While yielding three additional abundant heterotrophic phages, this method did not yield additional contigs containing any cyanophage marker genes (Fig. 1, Supplementary Data 3).

A previous report indicated that *Synechococcus* phage genomes occur in discrete clouds with a discontinuity in recruitment below ~95% identity[54]. While this pattern existed for some cyanophage reference genomes, having gaps in coverage at ~90–95% consistent with that idea, it was by no means the rule in our data for either those genomes or for the reference pelagiphage genomes (Figs. 2b, d, e).

Finally, the recruitment plots revealed a common pattern of high coverage of short regions within a genome or contig at up to 100% identity, whereas the rest of the genome or contig was only recruited at lower percent ID if at all (examples in Supplementary Figure 3); however some samples showed recruitment all along the same contig or genome (Supplementary Figure 3).

**T4-like cyanomyoviruses**. Few of the assembled viral contigs contained the myoviral marker gene g23 (major capsid protein), which was identified using an HMM (Hidden Markov Model) search of the predicted open reading frames (ORFs) (Supplementary Data 2). In order to determine whether cyanomyoviruses were actively infecting and simply did not assemble, we examined the unassembled reads, and assigned translated reads identified by the same Gp23-HMM to published protein sequences and assembled Gp23 ORFs of cyanomyoviruses and myoviruses infecting heterotrophic bacteria (see Supplementary Table 1 for accession numbers and Supplementary Figure 5 for a maximum likelihood tree of Gp23 proteins). Only two Gp23 ORFs from

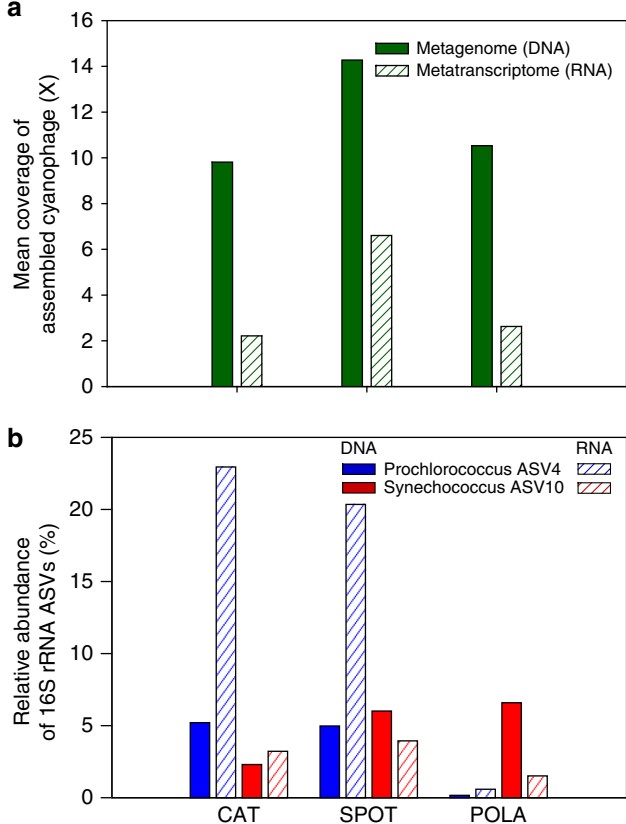

**Fig. 3** Matching assembled cyanophage with *Synechococcus* host using expression patterns. Presence and activity of the assembled cyanophage and its potential hosts in October 2012: **a** Mean coverage (after removing the top 25% most abundant contig positions) of the assembled cyanophage, **b** Amplicon single variants (ASVs) relative gene and transcript abundance of 16S-rRNA of the two most abundant cyanobacterial ASVs: *Prochlorococcus* ASV4 and *Synechococcus* ASV10. Note the near-absence of *Prochlorococcus* in POLA, in contrast to *Synechococcus* and the phage RNA, leading us to infer that the phage infected *Synechococcus*. Each group of bar plots, denoted by site name, represents data from a single sample

contigs cross-assembled with metagenomes were placed within cyanomyoviral clades. One had no recruitment in metatranscriptomes, implying that it was not actively infecting its host, and the other was persistent at SPOT but with low coverage (contigs 13 and 15 in Supplementary Data 1). Recruitment to Gp23 was generally higher at POLA except in January, and most of it was attributed to T4-like heterophages (phages infecting heterotrophic bacteria) rather than cyanophages. Published reference cyanophage proteins accounted for about 10–20% of total Gp23 transcripts, and there was minimal recruitment to assembled putative cyanomyoviruses, most of it at POLA in July (Supplementary Figure 6). Thus, it appears that cyanomyovirus genomes did not assemble.

**Cyanophage host matching**. One assembled contig that represented with high certainty, a partial cyanophage genome, was a putative cyanopodovirus (T7-like). This contig contained genes coding for photosystem-II protein D1 (*psbA*) and high-light induced protein (*hli*) that are reportedly widespread in cyanophages[28]. The putative cyanophage represented by this contig was actively transcribed (presumably infecting its host) in all three sites only in October 2012 (Fig. 3a). The cyanobacterial community by 16S-rRNA was dominated in October by two single amplicon variants (ASVs): one *Synechococcus* and one

*Prochlorococcus*. Both ASVs were present at SPOT and CAT in October, but only *Synechococcus* was present at POLA (Fig. 3b). Thus, we propose that this assembled contig represents a phage that infects *Synechococcus* ASV 10, which is 100% identical over the V4-V5 hypervariable region of 16S-rRNA to *Synechococcus* CC9902 of clade IV. On a phylogenetic tree of PS-II D1 proteins, the translated PS-II D1 of this phage clustered closely with a different cyanopodovirus isolated on *Synechococccus* (Supplementary Figure 7).

**Expression of viral and bacterial *psbA*.** The presence of *psbA* within the assembled cyanophage prompted us to survey expression of this gene in unassembled reads, which revealed comparable expression by cyanophages and cyanobacteria even as the relative abundance of *psbA* out of the entire metatranscriptome varied between samples (Supplementary Figure 8). Sharon et al.[44] previously showed that viral *psbA* gene variants can outnumber cyanobacterial *psbA* genes in metagenomes from the Mediterranean, and that viral gene expression is evident. We extended this to quantitatively partition gene expression into bacterial contribution from *Synechococcus* and *Prochlorococcus* and viral contribution from cyanomyoviruses and cyanopodoviruses, as evident from placing translated reads onto our PS-II D1 phylogenetic tree (Supplementary Figure 7). We found that *psbA* transcripts of T4-like cyanomyovirus origin generally accounted for roughly 50% ($51 \pm 10\%$) of cyanobacterial and cyanophage *psbA*, at a ratio of $1.2 \pm 0.6$ (mean ± standard deviation) (Fig. 4b). On several occasions, the viral variant exceeded the cyanobacterial variant in recruited read counts (Fig. 4).

In both metagenomes and metatranscriptomes, there was minor consistent recruitment to T7-like cyanopodovirus *psbA*. However, in every sample the contribution of T7-like cyanopodoviruses was very low compared to that of T4-like cyanomyoviruses.

## Discussion

Our determination of persistent vs. ephemeral infections depended on the ability to track read recruitment to contigs, hence it refers only to those viruses represented in contigs. While most of the active infections by viruses we assembled here were ephemeral, the minority that were persistent appeared in the more oligotrophic sites (CAT and SPOT) and no virus was persistently infecting its host at the heavily impacted POLA. This spatial distinction was probably due to varying anthropogenic and riverine inputs at POLA, which do not affect the other sites and can cause large, stochastic shifts in the microbial community of the port leading to differences in phage communities. Moniruzzaman et al.[55] also recently demonstrated dominance of ephemeral dynamics in infections of marine single-cell eukaryotes during an algal bloom, which is a drastically dynamic system. Conversely, Aylward et al. showed the occurrence of many persistent viruses in the relatively stable system of the Hawaii Ocean Time-series[26]. As the San Pedro Channel is a dynamic, seasonal system[56], it is not surprising that infections varied somewhat over time. An important consideration in determining persistence, which relies on read recruitment to contigs, is the extent that assembled contigs represent the entire viral community. Our results suggest that many viruses may be missed because they are not well represented in the contigs we were able to generate. Recruitment of reads to our assembled viral contigs and to published genomes of cyanophages and pelagiphages indicated that the assemblies originated from viruses with low microdiversity, whereas pelagiphages and cyanophages revealed genomic continuity reflecting high microdiversity. Due to high variation interfering with creation of contigs, this microdiversity probably hindered contig

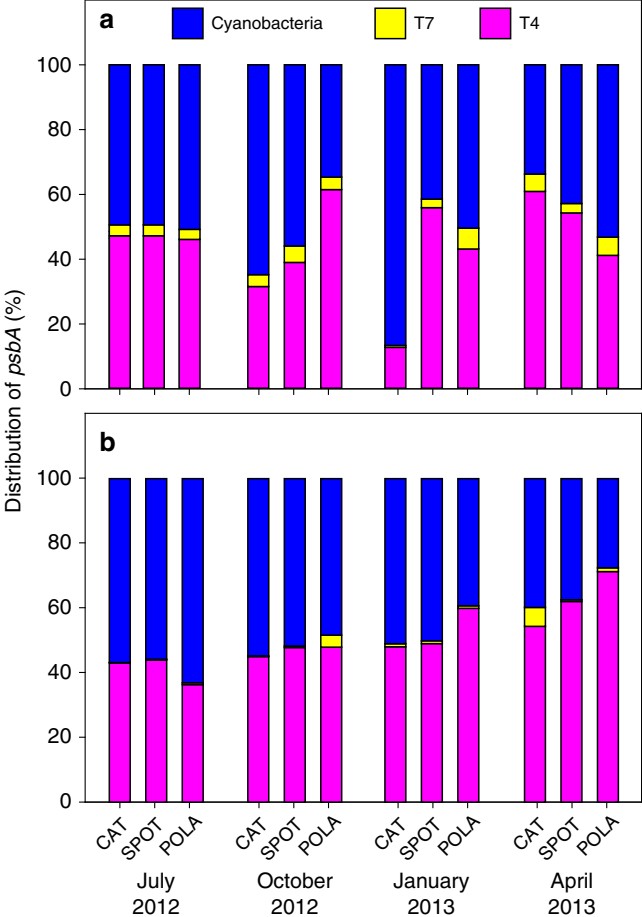

**Fig. 4** Phage and cyanobacteria expression of the PS-II reaction center. Relative abundance of *psbA* of cyanobacteria (*Synechococcus* and *Prochlorococcus*), T4-like cyanophages, and T7-like cyanophages out of total *psbA* in **a** metagenomes and **b** metatranscriptomes. Each stacked bar chart represents a single sample

dsDNA viruses in the ocean[57]. T4-like phages can comprise up to 25% of total virus-like particles[58] and the majority of total plaque-forming cyanophages[59]. Additionally, we know that the San Pedro Channel supports a diverse community of myoviruses and cyanobacteria[27,60]. Some of the problem could be high microdiversity 'breaking' the assemblies as noted above. That being said, there is some evidence suggesting that sampling time of day may be a factor. Cyanophage infections were shown to peak in the late afternoon in the central Pacific Ocean[26]. This is a reasonable mechanism considering that energy harvesting in the hosts is performed during the day, and then potentially exploited by the cyanophages[39,40,43,61]. This idea becomes even more plausible when discussing phages that carry photosystem genes, as there would be no fitness advantage to keeping those genes in the viral genome if infection did not take place in daylight. These genes are widely distributed in cyanomyoviruses[62], implying that in those phages the latent period would be especially likely to coincide with daylight. As our samples were collected early in the morning, it is possible that we were not able to assemble genomes of cyanomyoviruses from the metatranscriptomes because they were in the early phase of the lytic cycle and most of the genomes were not yet transcribed. Low recruitment to T4-like cyanomyovirus Gp23 protein compared to T4-like heterophages indicates that at the time of sampling few cyanophages were at the late phase of infection in which capsid protein genes are expressed.

Another theory on the timing of infection by cyanomyoviruses is that late infection coincides with the S/G2 phases in the host cell cycle, in which the host genome was already replicated but the cell hasn't divided yet, the logic being that there would be double the amount of nucleotides available for synthesis of phage genomes[63]. This is indeed the case in the T4 phage of *Escherichia coli*[64]. Evidence from cultures[65,66] and from environmental samples[67] indicate that for at least some strains of *Prochlorococcus* the host cells divide at dawn, which would imply that the bulk of phage transcription occurs during the night, and may also explain why we were unable to assemble cyanomyovirus genomes from our metatranscriptomes.

Similar theoretical arguments can be made regarding pelagiphages, which depend on light-energy harvesting using proteorhodopsin[68,69]. Indeed, at least one pelagiphage was previously shown to have diel variation of transcription[26]. In an attempt to circumvent the issue with cyanophage and pelagiphage assembly we recruited unassembled metagenomic and metatranscriptomics reads to curated cyanophage and pelagiphage genomes found in the RefSeq database. While the exact published genomes themselves were not present in our samples, we posit that other T4-like cyanophages closely-related to those published were present and persistently infecting their hosts. The broad recruitment covering a wide range of sequence identity to those genomes implies high genomic diversity of these phages. With proper sampling timing, and as assembly methods continue to improve, assembly of organisms with high-microdiversity may become more computationally manageable over multiple samples, and might reveal the exact cyanophage genomes present in samples like ours. Still, the pattern of a genomic continuum in cyanophages and pelagiphages, i.e. with few breakpoints or gaps in genomic identity between many close relatives, should remain relevant and informative.

Matching viral contigs and hosts is an ongoing challenge, but we were able to use genetic information and regional abundance patterns to make a likely match between a cyanopodovirus and a strain of *Synechococcus*. Here we take advantage of presence of the gene *psbA* in the assembled cyanopodovirus contig. This gene was presumably horizontally transferred to viral ancestors of cyanophages from their hosts. Despite its history of horizontal transfers, *psbA* is considered a good indicator of host genus[19].

assembly from local representatives of those groups[33]. Sampling time of day may have been another significant hurdle for cyanophages (see below). Regardless of the reason, lack of assembled cyanophages and pelagiphages may have contributed to the low fraction of traceable persistent infections out of all active infections, as cyanobacteria and the SAR11 clade consistently make up a significant portion of the prokaryotic community at SPOT and CAT, and therefore their phages are likely to be persistently infecting them as well. Many recruitment plots contained a pattern of coverage of a short sequence across a nucleotide identity range of 70–100%. This pattern highlights two issues: (1) some genes are so conserved or so often laterally transferred that their partial sequences cannot be used to identify whether a certain phage is present and (2) that mean coverage of contigs could be highly biased by these conserved regions, which needs to be considered when evaluating abundance of the contigs and for coverage-based binning of genomes. As an example of this consideration, we ignored the 25% most highly-recruited positions within each contig when calculating the contig mean coverage. We also note that widely-used recruitment algorithms (e.g. bowtie, bwa) only map reads with a local or end-to-end match at a very high percent identity and would therefore miss moderately close relatives to the query sequence that may be relevant to questions about phage ecology. We were surprised not to find multiple and abundant cyanomyoviral contigs, because such cyanophages are reported to be some of the most common

Additionally, cyanopodoviruses tend to have a narrow host range compared to cyanomyoviruses[14,19]. Therefore, it is somewhat easier to use community dynamics to match a host to a podovirus.

Statistical correlations between environmental bacteria and viruses are usually hard to interpret, as both positive and negative time-delayed correlations could indicate a host-phage relationship, depending on time scales. Demonstrating an active infection via transcription simplifies the issue, as an actively infecting phage must have host cells present at the time of sampling.

Many cyanophages contain a variety of genes that maintain photosynthetic activity in the host during infection, from spare parts for photosynthetic reaction centers through regulation and optimization of those apparati[70]. In particular, viruses were shown to maintain photosystem II function during infection under light conditions in order to maintain continuous supply of energy to the host, as transcription of host genes drops during infection and PS-II proteins have a short lifetime[13,37,39–41,61]. Phages that contain the *psbA* gene probably derive fitness advantages from it compared to phages that do not[34,43].

We show here that expression of viral *psbA* was comparable to cyanobacterial *psbA* year-round. This information can be used to roughly estimate the proportion of infected cyanobacteria from our *psbA* data and to compare it to previously published infection rates. During host infection, the number of phage mRNA molecules of *psbA* increases quickly early in the latent period of infection and becomes the main source of *psbA* transcripts in the cell[8,39,40,43,61]. Under the assumption that the average *psbA* expression is comparable in infected (viral origin) and uninfected (bacterial origin) cells, we can use the viral expression of *psbA* vs. the bacterial expression to give a rough estimate of the fraction of cyanobacterial cells infected with cyanophages. What we observe in the sample is a comparable expression of T4-like *psbA* and cyanobacterial *psbA*, which suggests that on average roughly half of the cyanobacteria are infected. This is in accordance with the high end of published estimates for marine cyanobacteria[4,57,71,72], and admittedly is very rough. But it confirms the idea that infection is an important part of cyanobacterial ecology. If indeed the lack of assembled cyanomyovirus genomes resulted from sampling in the early phase of lytic infection, we may even be underestimating viral *psbA* expression.

Gene abundance and expression of viral *psbA* of T4-like origin was always much higher than T7-like *psbA* in our samples. This may represent a real higher abundance of actively infecting but not assembled T4-like cyanophages. *psbA* is expressed throughout the lytic phase, including during early infection, which could account for its high expression despite the lack of cyanophage genome assembly. Other contributing factors may include the more specific host range reported for cyanopodoviruses compared to cyanomyoviruses[14,73,74] or that only clade B of T7-like cyanophages carries the *psbA* gene as opposed to nearly all T4-like cyanophages[19,62,73].

As no temperate marine T4-like and T7-like cyanophages have been reported, suggesting they are generally lytic[75,76], presence of their *psbA* genes in our metagenomes may result from viral genomic DNA in cells at the later stages of infection, pseudolysogeny, phages that adsorbed to cells or particles, or any free viruses incidentally caught on the filter. In other systems, lysogeny may account for contig-wide recruitment in metagenomes but not in metatranscriptomes.

Extending metatranscriptomic methods as recently applied to marine eukaryotic[55,77,78] and prokaryotic[26] viral infection, we show the power of multiple approaches to track viral infection and dynamics within the broad picoplankton community, using metatranscriptomes of the cellular fraction, with particular examples in the cyanobacteria. Read recruitment is an excellent way to track particular viruses, but this requires genomes or large contigs and the vast majority of viruses have not been isolated, nor genomes sequenced. While ephemeral and more clonal viral genomes assemble relatively easily from metatranscriptomes, and thus can be tracked by read recruitment, the more common and persistent viruses with many close relatives assembled poorly. For these, the use of marker genes is especially important. The observed infection dynamics can sometimes be used in combination with microbial community structure and viral marker genes to deduce a host. Finally, use of metagenomes and metatranscriptomes provides an insight into quantifiable viral contribution to photosynthesis.

## Methods

**Sample collection**. Surface seawater was collected by bucket on 15 July 2012, 19 October 2012, 9 January 2013, and 24 April 2013 in three locations: The Port of Los Angeles (33°42.75′N 118°15.55′W), the San Pedro Ocean Time-series (33°33.00′N 118°24.01′W), and Two Harbors, Santa Catalina Island (33°27.18′N 118°28.51′W). Water was collected between 7 am and 12 noon. Duplicate samples of 20 l were filtered in each location through an 80 μm mesh followed by a glass fiber syringe prefilter (Gelman, 4523) which collected the >1 μm size fraction and a 0.2 μm PES Sterivex filter (Millipore, SVGPB1010), which collected the free-living size fraction. The duration of filtration was 15–20 min, immediately after which RNAlater (Thermo-Fisher, AM7020) was added to each filter and filters were capped and flash frozen no more than 5 min post-filtration.

Viral and cellular counts as well as heterotrophic production were measured on the same day. For detailed methods please see Supplementary Methods.

**Library preparation**. RNAlater was removed from all filters prior to nucleic acid extraction with a syringe. DNA and RNA were extracted simultaneously from Sterivex filters by bead-beating using 0.1 mm glass beads added into the sterivex shell, followed by an AllPrep kit (Qiagen, 80204). An internal standard (ERCC RNA Spike-In Mix, Thermo-Fisher 4456740) was added into the lysate after bead-beating for quality assurance. RNA was enriched for mRNA with RiboZero (Illumina, MRZB12424). Resulting mRNA was reverse transcribed using SuperScript-III (Invitrogen, 18080–051). DNA and cDNA were physically sheared with Covaris m2 and size-selected for products larger than 300 bp with magnetic AMPure beads (Beckman-Coulter, A63881) at a ratio of 0.7 beads to product. RNA libraries were prepared and barcoded using NEBNext Ultra Directional RNA library Prep Kit for Illumina (New England Biolabs, E74205). DNA libraries were prepared and barcoded with Ovation UltraLow Library Prep V2 (Nugen, 0344). Metagenomes were sequenced on Illumina HiSeq 2 × 125 bp or 2 × 150 bp. Metatranscriptomes were sequenced on Illumina HiSeq 2 × 250 bp.

**Read processing and assembly**. Raw metagenomic and metatranscriptomic reads were quality trimmed, and residual ribosomal reads as well as the internal standard were removed informatically. Merged reads from each sample separately were assembled via several methods with or without subsampling (see Supplementary Methods). Only 1455 contigs larger than 5 kbp were further analyzed.

**Identification and annotation of viral contigs**. Viral contigs were identified by running VirSorter version 1.0.3[50] using RefSeq on the CyVerse platform on all contigs >5 kbp and only contigs classified as category 1 or category 2 were considered. VirSorter is a well-established and reliable tool, but it has an inherent database bias and performs better on longer contigs because they are more likely to contain hallmark viral genes. Thus, all assembled contigs >5 kbp (regardless of VirSorter results) were also ranked using VirFinder[51], which relies on k-mer signatures, and only contigs ranking higher than 0.85 were considered further. ORFs were predicted and annotated within those contigs, and hit taxonomy was used toward assigning a contig as viral or non-viral.

Read mapping to the viral contigs was used to describe temporal and spatial dynamics. For a detailed description please see Supplementary Methods.

**Microbial community composition analysis**. The V4-V5 regions of the 16S-rRNA coding gene were amplified from DNA and cDNA from all samples using the 515(N)-F and 926-R primers, and sequenced on an Illumina MiSeq 2 × 300 bp (UC Davis genome center) along with a negative controls and even and staggered mock communities. Reads were quality-trimmed, assembled and processed through the minimum entropy decomposition (MED) pipeline to produce amplicon single variants (ASVs) and calculate beta diversity (Supplementary Methods).

**Analysis of PS-II D1 protein sequences**. We built a maximum likelihood tree from a set of curated PS-II D1 protein sequences and placed the translated ORFs from our assembled cyanophages in it. The same set was used to build a hidden

Markov model (HMM) of PS-II D1 to which we mapped reads from all metagenomes and metatranscriptomes (see Supplementary Methods).

**Analysis of Gp23 protein sequences**. Metatranscriptomic and metagenomics reads were searched with blastx against a set of T4-like clusters of orthologous groups (COGs) with an E-value threshold of $10^{-5}$. Metatranscriptomic reads of 89,768 and 134,995 metagenomic reads were annotated as Gp23. An HMM and a maximum-likelihood tree of Gp23 were built as described in the Supplementary Methods for PS-II D1.

**Recruitment to phage genomes**. The four currently available full pelagiphage genomes were downloaded from NCBI and concatenated with assembled viral contigs from metatranscriptomes the metagenomes as well as with published cyanophage genomes downloaded from NCBI RefSeq (accession numbers in Supplementary Figure 1). Metagenome and metatranscriptomics reads were searched against this dataset with blastn using default settings. For metagenomes only hits longer than 100 bp were retained, and for metatranscriptomes only hits longer than 200 bp. Hits were then plotted against the genomes using R core package.

**Reporting summary**. Further information on experimental design is available in the Nature Research Reporting Summary linked to this article.

## Data availability

All raw data can be found on EMBL-ENA under project number PRJEB12234. Raw metatranscriptomics sequences accession numbers are ERS1864892-ERS1864903, and negative control library sequences accession number is ERR2089009. Raw metagenomic sequences accession numbers are ERS1869885-ERS1869896 and negative control accession number is ERS1872073. The 69 assembled viral contigs can be found in Genbank under Bioproject PRJNA472807 accession numbers QKOA01000001-QKOA01000069.

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

## Acknowledgements

The authors would like to thank R. Sachdeva, N. Ahlgren, A. Parada, L. Berdjeb, E. Graham, M. Lee, J. Ren, F. Sun and T. Delmont for insightful discussions and advice on bioinformatic analyses. We thank C. Roney-Garcia, the Sundiver crew and the USC Wrigley Institute of Environmental Studies for logistic support. This work was supported by NSF grant 1136818, Gordon and Betty Moore Foundation Marine Microbiology Initiative grant GBMF3779 and Norma and Jerol Sonosky summer fellowship to E.T.S.

## Author contributions

E.T.S. participated in work design, data acquisition, performed data analysis and interpretation and wrote and revised the manuscript. J.C.I.-E. Contributed to data analysis and interpretation and reviewed the manuscript. D.M.N. assisted in data acquisition and interpretation and reviewed the manuscript. E.B.F. assisted in data acquisition and analysis and reviewed the manuscript. J.A.F. Conceived the project, designed and supervised the work, assisted in interpretation and reviewed the manuscript.

## Additional information

**Competing interests:** The authors declare no competing interests.

