## [Peer Review File · Nature Communications]

This manuscript has been previously reviewed at another journal that is not operating a transparent peer review scheme. This document only contains reviewer comments and rebuttal letters for versions considered at Nature Communications.”

Reviewers' Comments:

Reviewer #1:

Remarks to the Author:

This is a much-improved version of the manuscript. The original manuscript had important concepts and data, but lacked clarity and had a number of clerical errors (missing figures, incorrect Figure numbering). Those issues have been addressed in this revised manuscript. In addition, all reviewers brought up important points about statements in the manuscript that lacked evidence but were compelling and interesting ideas. These statements have either been removed or revised appropriately.

Based on the revisions, I think this manuscript of sufficient quality for publication in Nature Communications.

REVIEWERS' COMMENTS:

Reviewer #1 (Remarks to the Author):

This is a much-improved version of the manuscript. The original manuscript had important concepts and data, but lacked clarity and had a number of clerical errors (missing figures, incorrect Figure numbering). Those issues have been addressed in this revised manuscript. In addition, all reviewers brought up important points about statements in the manuscript that lacked evidence but were compelling and interesting ideas. These statements have either been removed or revised appropriately.

Based on the revisions, I think this manuscript of sufficient quality for publication in Nature Communications.

We thank the reviewer for revisiting this manuscript and for their kind words.